# EXPRESSIVITY OF SHALLOW NEURAL NETWORKS OVER FINITE FIELDS

## ABSTRACT

We study the expressivity of shallow polynomial neural networks (PNNs) with monomial activation functions over finite fields. For a given architecture, we define a neuromanifold as the image of the map from all possible network weights into the product of polynomial rings. We quantify the expressivity by the cardinality of the neuromanifold, and derive a natural lower and upper bound. This leads to counting rational points over finite fields, a problem closely linked to the Weil conjectures. Finally, we present an architecture that exhibits a striking difference in the neuromanifolds when considered over a characteristic zero versus a finite-characteristic field, illustrating the critical role of field characteristic on the notion of expressivity.

## 1 INTRODUCTION

Neural networks have excelled in wide-ranging fields from computer vision to natural language processing in recent years. The capacity of a neural network architecture to represent a class of functions is referred to as its *expressivity* (Gühring et al., 2023). Neural networks dynamics and training are typically studied using statistical and probabilistic methods.

However, algebro-geometric approaches to neural networks have gained increasing attention in recent years and have been applied to diverse architectures. The direction of *neuroalgebraic geometry* (Marchetti et al., 2025) investigates networks through the lens of algebraic geometry, focusing on the properties of the functional space of their outputs, known as the *neuromanifold* (Kileel et al., 2019).

In this work, we study neuromanifolds over finite fields. This is motivated by the practical advantages of weight quantization, which reduces both energy consumption and storage requirements (Yuan & Agaian, 2023). Moreover, counting rational points over finite fields connects to the deep theory of the Weil conjectures. This allows one to compute the Betti numbers of the Zariski closure of the corresponding neuromanifolds over complex numbers (Baez, 2025).

In particular, we consider shallow neural networks over finite fields with monomial activation functions $\sigma(x) = x^r$, where the network output is a $k$-tuple of homogeneous polynomials of degree $r$. We analyze the neural network architecture via the associated *parameter map*, defined as the map from the space of all network parameters to the $k$-fold Cartesian product of a polynomial ring over a finite field.

Since the space of polynomials of bounded degree over a finite field is itself finite, we can quantify the expressivity of a network by computing the cardinality of its neuromanifold. If the neuromanifold coincides with the entire space, we say it is *filling* (Kubjas et al., 2024). The behavior of neuromanifolds over finite fields can differ drastically from their real or complex counterparts, even for small architectures. For instance, for the architecture $\mathbf{d} = (2, 2, 2)$ with $r = 2$, the neuromanifold over finite fields has cardinality approximately half that of the ambient space, whereas over the complex numbers the Zariski closure of the neuromanifold fills the entire ambient space.

### 1.1 RELATED WORK

The geometric structure of neural network function spaces was first investigated in Amari (1994), where the term *neuromanifold* was introduced to denote the function space associated with a given

architecture. A comprehensive survey of neural network expressivity is provided in Gühring et al. (2023).

The expressivity of polynomial neural networks (PNNs) was first analyzed in Kileel et al. (2019) through the notions of filling and thick architectures. The geometry of neuromanifolds and neurovarieties was subsequently developed in Kubjas et al. (2024), while the expressive power of PNNs was further investigated in Finkel et al. (2025). Questions of identifiability were addressed in Usevich et al. (2025), and a detailed study of singularities was presented in Shahverdi et al. (2025). In practice, PNNs have been shown to achieve performance comparable to networks with non-polynomial activation functions (Yavartanoo et al., 2021; Chrysos et al., 2022).

To improve computational efficiency and reduce storage and energy consumption, neural networks with low-precision weights have attracted significant attention (Courbariaux et al., 2014). Related directions include networks with rational coefficients (Averkov et al., 2025) and integer coefficients (Wu et al., 2018). The expressive capabilities of neural networks with limited-range integer weights were analyzed in Draghici (2002). Neural networks that operate exclusively with integers have been studied in Wang et al. (2022).

Significant attention has been devoted to binary and ternary neural networks, where weights are restricted to $0, 1$ (1-bit) and $-1, 0, 1$, respectively. The training of binarized neural networks with 1-bit weights and activations was first introduced in Hubara et al. (2016). Subsequently, ternary weight networks, offering improved efficiency over their binary counterparts, were proposed in Liu et al. (2023).

An extensive survey of neural network quantization was provided in Gholami et al. (2022). A detailed review of binary neural networks, covering both activations and weights and emphasizing hardware training advantages, was presented in Yuan & Agaian (2023).

Recent developments in large language model (LLM) architectures have shown that weight quantization can be applied with minimal loss in performance, while yielding substantial gains in energy efficiency and storage requirements (Ma et al., 2024). A comprehensive study of quantization techniques for LLMs was presented in Jin et al. (2024).

## 1.2 MAJOR CONTRIBUTIONS

Our main contributions are summarized as follows:

1. We formulate the expressivity of polynomial neural networks over finite fields in terms of the cardinality of its neuromanifold $\mathcal{P}_{\mathbf{d},r}$.

2. We compare shallow networks over finite fields with their complex counterparts, highlighting the key differences between them.

3. We compute the cardinalities of neuromanifolds for various architectures and establish a general upper and lower bound.

A summary of the computed architectures is presented in Table 1 in Section 1.3.

## 1.3 PAPER OUTLINE

In Section 2, we introduce the notion of a shallow neural network over finite fields, its ambient space, and establish a general upper bound on the cardinality of neuromanifolds. We then present results for several shallow neural network architectures, including:

- Section 3: architectures $\mathbf{d} = (n, m, k)$ with square activation $\sigma(x) = x^2$ over a finite field $\mathbb{F}$ of characteristic $\neq 2$, and a lower bound on architectures $\mathbf{d} = (n, n, k)$ with arbitrary monomial activation $\sigma(x) = x^r$,

- Section 4: architectures $\mathbf{d} = (n, 1, k)$ and $\mathbf{d} = (n, 2, k)$ with arbitrary monomial activation $\sigma(x) = x^r$ over a finite field $\mathbb{F}$ of arbitrary characteristic,

- Section 5: architectures $\mathbf{d} = (n, m, k)$ with activation degree $r$ divisible by the characteristic $p$ of the field $\mathbb{F}$.

Finally, Section 6 concludes with remarks and a summary of the results.

Table 1: Main Results

| $\mathbf{d}$ | $r$ | $p$ | $|\mathcal{P}_{\mathbf{d},r}|$ | Reference |
|---|---|---|---|---|
| $(n,m,1)$ | $2$ | $p \neq 2$ | $\min\left(\sum_{s=0}^{m} N(n,s),\, q^{\binom{n+r-1}{r}}\right)$ | 3.3.1 |
| $(n,1,k)$ | any | any | $\dfrac{(q^n-1)(q^k-1)}{(q-1)}+1$ | 4.2 |
| $(n,2,k)$ | $\neq p^i$ | any | $\begin{aligned}&(q-1)\left|\overline{\mathcal{P}_{(n,2,1),r}}\right|\left|\mathbb{P}^{k-1}\right| \\ &+\binom{\left|\overline{\mathcal{P}_{(n,1,1),r}}\right|}{2}N_2(2,k)+1\end{aligned}$ | 4.5.1 |
| $(n,2,k)$ | $p^i$ | any | $\begin{aligned}&(q-1)\left|\overline{\mathcal{P}_{(n,2,1),r}}\right|\left|\mathbb{P}^{k-1}\right| \\ &+\dfrac{2}{q(q+1)}\binom{\left|\overline{\mathcal{P}_{((n,1,1),r)}}\right|}{2}N_2(2,k)+1\end{aligned}$ | 4.5.1 |
| $(n,m,k)$ | $ip$ | any | $|\mathcal{P}_{\mathbf{d},i}|$ | 5.1 |
| $(n,m,k)$ | $p^i$ | any | $\min\left(\sum_{i=1}^{m} N_i(n,k)+1,\, q^{kn}\right)$ | 5.2 |
| Upperbound: | | | | |
| $(n,m,k)$ | any | any | $|\mathcal{P}_{(n,m,k),r}| \leq p^{\gamma_{n,p}(r)\cdot k}$ | 2.2 |

It is going to be useful to mention the formulas that we will use across the paper below:

- A $q$-binomial coefficient is equal to

$$\begin{bmatrix} a \\ b \end{bmatrix}_q = \frac{(1-q^a)(1-q^{a-1})\cdots(1-q^{a-b+1})}{(1-q)(1-q^2)\cdots(1-q^b)} \tag{1}$$

- In (Conrad, 1995), if $r = \sum_{t=0}^{s_r} r_t p^t$ is the base$-p$ expansion of the activation degree, then

$$\gamma_{n,p}(r) = \prod_{t=0}^{s_r} \binom{r_t+n-1}{r_t}. \tag{2}$$

- According to Morrison (2006, Section 1.7), the number of $k \times n$ matrices of rank $m$ over a finite field $\mathbb{F}$ with $q$ elements is

$$N_m(n,k) = \begin{bmatrix} k \\ m \end{bmatrix}_q (q^n-1)(q^n-q)\ldots(q^n-q^{m-1}). \tag{3}$$

- According to MacWilliams (1969, Section 2) the number of symmetric $n \times n$ matrices of rank $m$ is

$$N(n,m) = \prod_{i=1}^{\lfloor \frac{m}{2} \rfloor} \frac{q^{2i}}{q^{2i}-1} \cdot \prod_{i=0}^{m-1}(q^{n-i}-1). \tag{4}$$

## 2 PRELIMINARY

### 2.1 NEURAL NETWORK DEFINITION

Let $\mathbb{F}$ be a field, and fix a triple of natural numbers $\mathbf{d} = (n, m, k)$. Let $\sigma : \mathbb{F} \to \mathbb{F}$ be the *monomial activation function* with the activation degree $r \in \mathbb{N}$, that is, $\sigma(x) = x^r$. A *shallow neural network* $f_{\mathbf{w}} : \mathbb{F}^n \to \mathbb{F}^k$ with architecture $\mathbf{w} = (\mathbf{d}, \sigma)$ is the composition

$$\mathbb{F}^n \xrightarrow{W_1} \mathbb{F}^m \xrightarrow{\sigma_1} \mathbb{F}^m \xrightarrow{W_2} \mathbb{F}^k,$$

where $W_1 \in \mathbb{F}^{m \times n}, W_2 \in \mathbb{F}^{k \times m}$ are matrices, and $\sigma_1 : \mathbb{F}^m \to \mathbb{F}^m$ is the coordinate-wise application of the activation function $\sigma$. Equivalently,

$$f_{\mathbf{w}}(\mathbf{x}) = W_2(\sigma_1(W_1 \mathbf{x})), \quad \mathbf{x} = \begin{bmatrix} x_1 & \dots & x_n \end{bmatrix}^T.$$

For the remainder of this paper, we denote the entries of $W_1$ and $W_2$ by $a_{ij}$ and $b_{ij}$, respectively. We denote the $i$th component of $f_{\mathbf{w}}$ by $f_{i,\mathbf{w}}$ and observe that it is a homogeneous polynomial of degree $r$ over $n$ variables of the form

$$f_{i,\mathbf{w}}(\mathbf{x}) = \sum_{s=1}^m b_{is}(a_{s1}x_1 + \cdots + a_{sn}x_n)^r. \tag{5}$$

### 2.2 ABOUT THE AMBIENT SPACE

Let $\mathcal{F}(\mathbb{F}^n, \mathbb{F}^k)$ be the space of all functions from $\mathbb{F}^n$ to $\mathbb{F}^k$. The *expressive power* of a neural network refers to the extent to which the neural network can approximate functions within the space $\mathcal{F}(\mathbb{F}^n, \mathbb{F}^k)$ (Gühring et al., 2023). If $\mathbb{F} = \mathbb{F}_q$ is a finite field, then the evaluation map

$$\begin{aligned} eval : \mathbb{F}_q[x_1, \dots, x_n] &\longrightarrow \mathcal{F}(\mathbb{F}_q^n, \mathbb{F}_q^k) \\ f(x) &\longmapsto [a \mapsto f(a)] \end{aligned} \tag{6}$$

is not injective. For example, if $\mathbb{F}_p$ is a prime field, i.e., $q = p$ is a prime number, then a zero polynomial $\mathbf{o}(x) = 0$ and the polynomial $f(x) = x^p - x$ both map to the zero function in $\mathcal{F}(\mathbb{F}_p^n, \mathbb{F}_p^k)$ as $x^p \equiv x \bmod p$.

A space in which a neural network $f_{\mathbf{w}}$ resides is known as an *ambient space*. In this work, as a first step, we propose to study the expressivity by considering neural networks with polynomial activation functions as elements of a polynomial ring. In other words, we take the ambient space to be a Cartesian product of polynomial rings. This suggests the following definition.

**Definition 2.1.** Polynomials $g_1, \dots, g_k \in \mathbb{F}[x_1, \dots, x_n]$ can be *expressed* by a neural network $f_{\mathbf{w}}$ with architecture $(\mathbf{d}, \sigma)$ if there exist weights $\mathbf{w} \in \mathbb{F}^{m(n+1)}$ such that $f_{i,\mathbf{w}}(\mathbf{x})$ and $g_i(\mathbf{x})$ are the same element in the polynomial ring $\mathbb{F}[x_1, \dots, x_n]$ for all $i$.

Let $S^r(\mathbb{F}^n)$ be the space of homogeneous polynomials of degree $r$ over $n$ variables. Since $f_{i,\mathbf{w}}$ is a homogeneous polynomial according to Equation 5, then we can shrink our ambient space for this network architecture to the product of $S^r(\mathbb{F}^n)$. We can identify the space $S^r(\mathbb{F}^n)$ with the vector space $\mathbb{F}^{\binom{n+r-1}{r}}$ where we take the coefficients of a homogeneous polynomial in the standard monomial basis.

We define the *parameter map*

$$\begin{aligned} \Psi_{\mathbf{d},r} : \mathbb{F}^{m(n+k)} &\to (S^r(\mathbb{F}^n))^k \\ \mathbf{w} &\mapsto f_{\mathbf{w}} := (f_{1,\mathbf{w}}, \dots, f_{k,\mathbf{w}}) \end{aligned} \tag{7}$$

**Definition 2.2.** The *neuromanifold*, denoted by $\mathcal{P}_{\mathbf{d},r}$, is the image of the parameter map $\Psi_{\mathbf{d},r}$.

**Example 2.1.** Let $d = (2, 2, 1), r = 2$, and $char(\mathbb{F}) \neq 2$. Then the output of the neural network is equal to

$$f_{\mathbf{w}}(\mathbf{x}) = W_2(\sigma_1(W_1 \mathbf{x})) = \begin{bmatrix} b_1 & b_2 \end{bmatrix} \sigma_1 \left( \begin{bmatrix} a_{11} & a_{12} \\ a_{21} & a_{22} \end{bmatrix} \begin{bmatrix} x_1 \\ x_2 \end{bmatrix} \right)$$

After a quick computation, we obtain

$$f_{\mathbf{w}}(\mathbf{x}) = (b_1 a_{11}^2 + b_2 a_{21}^2) x_1^2 + (2 b_1 a_{11} a_{12} + 2 b_2 a_{21} a_{22}) x_1 x_2 + (b_1 a_{12}^2 + b_2 a_{22}^2) x_2^2$$

Therefore, the parameter map $\Psi_{\mathbf{d},r} : \mathbb{F}^6 \to S^r(\mathbb{F}^2) \cong \mathbb{F}^3$ is given by

$$(a_{11}, a_{12}, a_{21}, a_{22}, b_1, b_2) \mapsto \left( b_1 a_{11}^2 + b_2 a_{21}^2, \; 2 b_1 a_{11} a_{12} + 2 b_2 a_{21} a_{22}, \; b_1 a_{12}^2 + b_2 a_{22}^2 \right)$$

A central problem in this context is to understand *the expressive capacity* of neural network architectures within the ambient space $(S^r(\mathbb{F}^n))^k$ (Kileel et al., 2019).

**Definition 2.3.** We measure the *expressive capacity* of the network by computing the ratio

$$|\mathcal{P}_{\mathbf{d},r}| / |\left(S^r(\mathbb{F}^n)\right)^k|.$$

### 2.3 Upper Bounds of Neuromanifolds

Using the multinomial congruence theorem (Conrad, 1995), we can find a natural upper bound for neuromanifolds over prime fields $\mathbb{F}_p$. For example, if $\mathbf{d} = (2, 1, 1), r = 2$, and $p = 2$, then

$$f_{\mathbf{w}}(\mathbf{x}) = b_{11}(a_{11} x_1 + a_{12} x_2)^2 = b_{11} a_{11}^2 x_1^2 + 2 b_{11} a_{11} a_{12} x_1 x_2 + b_{11} a_{12}^2 x_2^2 = b_{11} a_{11}^2 x_1^2 + b_{11} a_{12}^2 x_2^2.$$

In other words, the coefficient in front of $x_1 x_2$ is always zero as we take the coefficients in $\mathbb{F}_2$. The coefficients in front of $x_i^2$ can be either 0 or 1. This gives us that $|\mathcal{P}_{(2,1,1),2}| \leq 2^2$.

**Proposition 2.2.** If $\mathbf{d} = (n, m, k)$ and $\sigma(x) = x^r$ with $r = \sum_{t=0}^{s_r} r_t p^t$ over a prime field $\mathbb{F}_p$ and $\gamma_{n,p}(r)$ is as in Equation 2, the cardinality of the neuromanifold satisfies the following upper bound:

$$|\mathcal{P}_{(n,m,k),r}| \; \leq \; \left( p^{\gamma_{n,p}(r)} \right)^k.$$

*Proof.* See Appendix A.1. $\qquad\square$

## 3 Shallow networks with $r = 2$ and $char(\mathbb{F}) \neq 2$

The $i$th output $f_{i,\mathbf{w}}$ of the neural network corresponds to a symmetric tensor of order $r$, which we will denote by $A_i$. Then, the output of the neural network $f_{\mathbf{w}}$ can be identified with a $k$-tuple $(A_1, ..., A_k)^T$ of symmetric tensors, each satisfying the symmetric CP-decomposition

$$A_i := b_{i1} L_1^{\otimes r} + \cdots + b_{im} L_m^{\otimes r} \tag{8}$$

where each $L_j \in \mathbb{F}^n$ is $j$th row of $W_1$ (see Landsberg & Teitler (2010).

However, over finite fields, not every symmetric tensor has a well-defined CP-rank (see Friedland & Stawiska (2013, Proposition 7.1)). Therefore, there exist shallow architectures whose ambient space contains elements that cannot be expressed via the decomposition 8. But, in the case when $r = 2$ and $char(\mathbb{F}) \neq 2$, as we will see in the next section, every symmetric matrix has a well-defined CP-rank. This allows us to compute the cardinality of the neuromanifold for architectures $\mathbf{d} = (n, m, 1)$, and $r = 2$.

### 3.1 Single Output Architectures $\mathbf{d} = (n, m, 1)$ with $r = 2, char(\mathbb{F}) \neq 2$

For this architecture, the output of the neural network $f_{\mathbf{w}}$ according to Equation 8 is represented by a single $n \times n$ symmetric matrix of the form

$$A = b_1 L_1^\top L_1 + \cdots + b_m L_m^\top L_m = W_1^T D(W_{2,1}) W_1, \tag{9}$$

where $W_{2,i}$ is the $i$th row of $W_2$ and $D(\mathbf{v})$ is the diagonal matrix formed by placing the entries of a vector $\mathbf{v}$ on the diagonal. The theorem below (a reformulation of Axler (2015, Theorem 9.13)) shows that every symmetric matrix over any field of charecteristic $\neq 2$ has a well-defined CP-rank.

**Theorem 3.1** (Corollary of Axler (2015, Theorem 9.13)). *For every symmetric matrix $C$ over a finite field of characteristic $\neq 2$, there exists a diagonal matrix $D$ and an invertible matrix $Q$ such that $C = QDQ^\top$.*

This is a direct consequence of the classical linear algebra result that every symmetric bilinear form over a field of characteristic $\neq 2$ can be diagonalized.

According to Kubjas et al. (2024) (see Lemma 3.3),

$$\mathcal{P}_{(n,m,1),r=2} \text{ over } \mathbb{R} \text{ is equivalent to the set of symmetric } n \times n \text{ matrices of rank} \leq m \tag{10}$$

Theorem 3.1 tells us that the above description can be extended to any field of characteristic $p \neq 2$. Therefore, computing the cardinality of the neuromanifold $\mathcal{P}_{d,r}$ is equivalent to counting the number of $n \times n$ symmetric matrices of rank $\leq m$ over $\mathbb{F}_q$.

If $m \geq n$, then clearly $\mathcal{P}_{\mathbf{d},r}$ is the entire space of symmetric matrices $S^r(\mathbb{F}_q^n)$:

**Lemma 3.2.** *If* $\mathbf{d} = (n,m,1)$, $r = 2$, $p \neq 2$, *then*

$$\mathcal{P}_{d,r} = S^r(\mathbb{F}_q^n) \iff m \geq n$$

For the cases where $m \leq n$, we refer to a paper by MacWilliams (1969) which counts the number of $n \times n$ matrices of rank $m$ over $\mathbb{F}_q$.

**Theorem 3.3** (cf. MacWilliams (1969), Theorem 2). *Let* $N(n,m)$ *denote the number of* $n \times n$ *symmetric matrices of rank* $m$ *over a field* $\mathbb{F}_q$ *of characteristic not equal to 2. Then,* $N(n,m)$ *is given by Equation 4.*

As an immediate corollary,

**Corollary 3.3.1.** *For* $m \leq n$,

$$|\mathcal{P}_{(n,m,1),2}| = \sum_{j=0}^{m} N(n,j)$$

The next section considers architectures with output width $k = 2$.

## 3.2 Shallow Architectures $\mathbf{d} = (n,n,2), r = 2, char(\mathbb{F}) \neq 2$

In this section, we show that understanding the problem of simultaneous diagonalization of symmetric matrices over finite fields helps in understanding neuromanifold corresponding to the shallow network with multiple outputs. In particular, we restrict ourselves to studying the architectures $\mathbf{d} = (n,n,2)$ and show how the embeddings of the neuromanifolds over complex and finite fields into the ambient space are drastically different.

According to Equation 9, a point $(A_1, A_2)$ belongs to the neuromanifold $\mathcal{P}_{(n,n,2),2}$ if and only if

$$A_i = b_{i1} L_1^\top L_1 + \cdots + b_{im} L_m^\top L_m = W_1^T D(W_{2,i}) W_1 \tag{11}$$

for $i = 1, 2$. We are going to follow the definition of simultaneous diagonalization of symmetric matrices given in Bustamante et al. (2020).

**Definition 3.1.** A set of $k$ symmetric matrices $A_1, \ldots, A_k$ are *simultaneously diagonalizable via congruence* (SDC) if there exists an invertible matrix $Q \in GL_n(\mathbb{F})$ such that

$$A_i = QD_iQ^T, \quad D_i \text{ is diagonal for all } i = 1, ..., k.$$

It follows from the above definition that $A_1, A_2$ are simultaneously diagonalizable, then they lie in $\mathcal{P}_{(n,n,2),2}$. The opposite direction unfortunately does not hold as the matrix $Q$ may not be invertible. In other words, we can find a pair of symmetric matrices $(A_1, A_2)$ that satisfy decomposition equation 11, but are not simultaneously diagonalizable.

**Example 3.4.** If $\mathbf{d} = (3,3,2)$, then the network output for the forms $L_1(\mathbf{x}) = x_1$, $L_2(\mathbf{x}) = x_2$, and $L_3(\mathbf{x}) = x_1 + x_2$ equals to

$$f_{1,\mathbf{w}}(x_1, x_2, x_3) = b_{11}x_1^2 + b_{12}x_2^2 + b_{13}(x_1 + x_2)^2$$
$$f_{2,\mathbf{w}}(x_1, x_2, x_3) = b_{21}x_1^2 + b_{22}x_2^2 + b_{23}(x_1 + x_2)^2$$

Taking $b_{11} = b_{12} = -1, b_{13} = 1$ and $b_{21} = b_{23} = 0, b_{22} = 1$, we obtain $f_{1,\mathbf{w}}(\mathbf{x}) = 2x_1x_2$ and $f_{2,\mathbf{w}}(\mathbf{x}) = x_2^2$, which are not simultaneously diagonazible according to Lemma A.2.

By applying an algorithm for determining SDC given by Bustamante et al. (2020), we can partially extend the results on filling architectures for shallow networks over $\mathbb{C}$. This yields the following.

**Theorem 3.5.** *If $\mathbf{d} = (n, n, 2), r = 2$, and $\mathbb{F} = \mathbb{C}$, then the Zariski closure of $\mathcal{P}_{\mathbf{d},r}$ over $\mathbb{C}$ is the entire ambient space $S^2(\mathbb{C}^n) \times S^2(\mathbb{C}^n)$, but $\mathcal{P}_{\mathbf{d},r} \neq S^2(\mathbb{C}^n) \times S^2(\mathbb{C}^n)$.*

*Proof.* See Appendix A.2. $\qquad\square$

One might hope $\mathcal{P}_{(n,n,2),r}$ over $\mathbb{F}_q$ behaves similarly to its analogue over $\mathbb{C}$. Previously when $k = 1$, the neuromanifold $\mathcal{P}_{(n,n,1),r}$ fills the ambient space over both $\mathbb{C}$ and $\mathbb{F}_q$. However, their patterns diverge for $k > 1$.

For example, if $k = 2$, the neuromanifold over $\mathbb{C}$ contains an open Zariski set whose closure fills the entire ambient space. Contrastingly, the neuromanifold over a finite field $\mathbb{F}_q$ only fills in approximately half of the ambient space.

**Example 3.6.** $|\mathcal{P}_{(2,2,2),r=2}(\mathbb{F}_3)| = 393$, the ambient space has cardinality $|S^3(\mathbb{F}_q)|^2 = 729$, and their expressive power defined in Definition 2.3 is $\sim 0.539$. If we increase $q$ to 11, then $|\mathcal{P}_{(2,2,2),r=2}(\mathbb{F}_{11})| = 2415673, |S_2(\mathbb{F}_{11}^2)| = 4826809$, and their ratio is $\sim 0.500$.

**Conjecture 3.1.** The expressive capacity of $\mathcal{P}_{(2,2,2),r=2}$ over a prime field $\mathbb{F}_p$ approaches $\frac{1}{2}$ as $p \to \infty$.

We propose the following two open questions that can advance our understanding of neuromanifolds over finite fields with architectures where $k > 1$. The first concern is finding architectures whose expressive capacity converges to a non-zero fraction.

**Problem 3.7.** Find all architectures $\mathbf{d} = (n, m, k)$ over a prime field $\mathbb{F}_p$ whose expressive capacity converges to a non-zero fraction $0 < a/b < 1$ as prime $p \to \infty$.

The second question is a generalization of the problem of simultaneous diagonalization via congruence. The congruence action $Q \cdot C = QCQ^T$ on matrices generalizes to symmetric tensors of arbitrary order $r$ (see Grosdos et al. (2025), Remark 3.5)). For a change-of-basis $Q$ on $\mathbf{x}$ and a homogeneous form $C$,

$$(Q \cdot C)(\mathbf{x}) := C \circ (Q\mathbf{x})^{\otimes r}.$$

The tensor associated with $C$ is SDC if there exists a $Q$ for which $Q \cdot C$ is a diagonal tensor.

**Lemma 3.8.** *The set of SDC $k$-tuples of $n^{\otimes r}$ tensors is a subset of $\mathcal{P}_{(n,n,k),r}$.*

This follows from choosing $W_1$ to be invertible, and motivates our second question.

**Problem 3.9.** Given a tuple of symmetric tensors $(A_1, A_2, \ldots, A_k)$ over $\mathbb{C}$ or over $\mathbb{F}_q$, determine when they are simultaneously diagonalizable via congruence.

Problem 3.9 is solved geometrically in the case $\mathbf{d} = (3, 3, 2)$ and $r = 2$ by Kusejko (2016). In the case $\mathbf{d} = (n, m, k)$ and $r = 2$, Problem 3.9 was solved by Bustamante et al. (2020) over $\mathbb{C}$.

# 4 SHALLOW ARCHITECTURES $\mathbf{d} = (n, 1, k), (n, 2, k)$

In this section, we study neuromanifolds through the perspective of projective space. One of the main motivations for viewing neuromanifolds from this perspective is a connection to counting rational points of algebraic varities over a finite field. In particular, the point count allows one to compute local zeta function from the Weil conjectures (Baez, 2025). This gives information about the motivic structure of the Zariski closure of the complex analogue of the neuromanifold through its cohomology.

## 4.1 PROJECTIVE SPACE

Let $\mathbb{F}_q$ be a finite field. An affine space $\mathbb{F}_q^{n+1}$ is the $(n+1)$-fold Cartesian product of $\mathbb{F}_q$. Observe the cardinality $|\mathbb{F}_q^{n+1}| = q^{n+1}$. A projective space $\mathbb{P}^n(\mathbb{F}_q)$ is the set

$$\mathbb{P}^n(\mathbb{F}_q) := (\mathbb{F}_q^{n+1} - \{(0, \ldots, 0)\})/\sim$$

where $(x_0, \ldots, x_n) \sim (y_0, \ldots, y_n)$ if and only if there exists a non-zero scalar $\lambda$ such that $x_i = \lambda y_i$ for all $i$. A point in the projective space $\mathbb{P}^n(\mathbb{F}_q)$ will be indicated by $[x_0 : \cdots : x_n]$ where

$$[x_0 : x_1 : \cdots : x_n] := \{(\lambda x_0, \lambda x_1, \cdots, \lambda x_n) : \lambda \in (\mathbb{F}_q - \{0\})\}.$$

Let $\overline{\mathcal{P}_{\mathbf{d},r}}$ be the image of $\mathcal{P}_{\mathbf{d},r}$ in the projective space under the map

$$pr : \mathbb{F}_q^N \to \mathbb{P}^{N-1}(\mathbb{F}_q), \quad (x_1, \ldots, x_N) \mapsto [x_1 : \cdots : x_N]$$

where $N = \dim((S^r(\mathbb{F}^n))^k)$. The following lemma relates the cardinality of $\overline{\mathcal{P}_{\mathbf{d},r}}$ with the non-projective $\mathcal{P}_{\mathbf{d},r}$.

**Lemma 4.1.**

$$|\mathcal{P}_{\mathbf{d},r}| = (q-1)|\overline{\mathcal{P}_{\mathbf{d},r}}| + 1.$$

*Proof.* See A.3. $\square$

Geometrically, one can think of projective space as the set of all lines through the origin in $\mathbb{F}_q^{n+1}$. In relation to Equation 8, the decomposition of the $A_i$'s depends on the scalars $b_{ij}$ and the lines generated by $L_j$. Recall that each $L_j(\mathbf{x})$ is defined as the $j$th row of a vector $W_1\mathbf{x}$, i.e.,

$$L_j(\mathbf{x}) = a_{j1}x_1 + \cdots + a_{jn}x_n.$$

Considering $L_j$ as lines will simplify point-counting in proofs of results in the next section by eliminating some overcounting.

### 4.2 THE CASE $m = 1, 2$

Let $\mathbb{P}^k := \mathbb{P}^k(\mathbb{F}_q)$. For $m = 1$, we have the following count.

**Lemma 4.2.** *For $\mathbf{d} = (n, 1, k)$, any $r$ and any prime $p$, then*

$$|\overline{\mathcal{P}_{\mathbf{d},r}}| = |\overline{\mathcal{P}_{(n,1,1),r}}||\mathbb{P}^{k-1}| = \frac{(q^n-1)(q^k-1)}{(q-1)^2}.$$

*Proof.* See Appendix A.4. $\square$

For larger values of $m$, $\overline{\mathcal{P}_{\mathbf{d},r}}$ can be partitioned by the dimension of the span of the $k$-tuple $(A_1, \ldots, A_k)$ as a vector space over $\mathbb{F}_q$. We are thus able to describe $\overline{\mathcal{P}_{\mathbf{d},r}}$ in terms of spaces associated with CP-rank smaller than $m$, by studying how the rank-1 components combine linearly. For $m = 2$, this yields 12 and 13 below, depending on whether $r > 1$ or $r = 1$.

**Proposition 4.3.** *For $\mathbf{d} = (n, 2, k)$, $p \nmid r$, $r > 1$,*

$$\left|\overline{\mathcal{P}_{\mathbf{d},r}}\right| = \left|\overline{\mathcal{P}_{(n,2,1),r}}\right| \left|\mathbb{P}^{k-1}\right| + \binom{\left|\overline{\mathcal{P}_{(n,1,1),r}}\right|}{2} \frac{N_2(2,k)}{q-1} \quad (12)$$

*where $N_m(n, k)$ represents the number of $k \times n$ matrices of rank $m$ over $\mathbb{F}_q$ (see Equation 3).*

*Proof.* See Appendix A.5 $\square$

**Proposition 4.4.** *For $\mathbf{d} = (n, 2, k)$, $r = 1$,*

$$\left|\overline{\mathcal{P}_{\mathbf{d},r}}\right| = \left|\overline{\mathcal{P}_{(n,2,1),r}}\right| \left|\mathbb{P}^{k-1}\right| + \frac{2}{(q+1)q} \binom{\left|\overline{\mathcal{P}_{(n,1,1),r}}\right|}{2} \frac{N_2(2,k)}{q-1} \quad (13)$$

*Proof.* See Appendix A.6. $\square$

**Problem 4.5.** We imagine counting formulas of a similar vein exist for $m = 3, 4, 5...$, but the combinatorial complexity grows quickly, and thus, a full characterization remains open.

The next section will address the case $p \mid r$. In particular, Proposition 5.1 will state

$$|\mathcal{P}_{\mathbf{d},ip}| = |\mathcal{P}_{\mathbf{d},i}|, \quad (14)$$

which allows us to reduce architectures with $p \mid r$ to the situation $p \nmid r$ considered in the previous two propositions. As a first application of this identity, we obtain the following corollary.

**Corollary 4.5.1.** *For $\mathbf{d} = (n, 2, k)$, if $r \neq p^i$, $\left|\overline{\mathcal{P}_{\mathbf{d},r}}\right|$ satisfies Equation 12. If $r = p^i$, then $\left|\overline{\mathcal{P}_{\mathbf{d},r}}\right|$ satisfies Equation 13.*

## 5   SHALLOW ARCHITECTURES WITH $char(\mathbb{F}) \mid r$

In this section, we consider architectures where the activation degree $r$ of the activation function $\sigma$ is divisible by the prime characteristic of a finite field $F$. First, let us express the cardinality of the neuromanifold with $r = ip$ for some $i \in \mathbb{N}$ as the cardinality of the neuromanifold with $r = i$. In other words, we have the following result.

**Proposition 5.1.** If $\mathbf{d} = (n, m, k)$, $char(\mathbb{F}) = p$, $\sigma(x) = x^r$ where $r = ip$ for $i \in \mathbb{N}$, then

$$|\mathcal{P}_{\mathbf{d},ip}| = |\mathcal{P}_{\mathbf{d},i}|.$$

*Proof.* See Appendix A.7.  □

**Corollary 5.1.1.** If $\mathbf{d} = (n, m, k)$, $char(\mathbb{F}) = p$, $\sigma(x) = x^r$ where $r = sp^\ell$ for $\ell, s \in \mathbb{N}$ and $gcd(p, s) = 1$, then

$$|\mathcal{P}_{\mathbf{d},r}| = |\mathcal{P}_{\mathbf{d},s}|.$$

*Proof.* See Appendix A.8.  □

The case $s = 1$, i.e., $r = p^\ell$ for some positive integer $\ell$, can be computed explicitly as shown below.

**Proposition 5.2.** If $\mathbf{d} = (n, m, k)$, $char(\mathbb{F}) = p$, and $\sigma(x) = x^r$ with $r = p^\ell$ for some positive integer $\ell$, then $|\mathcal{P}_{\mathbf{d},r}|$ is precisely the number of $k \times n$ matrices over $\mathbb{F}$ of rank at most $m$.

*Proof.* See Appendix A.9.  □

As a consequence, we can arrive at an explicit expression for the expressivity.

**Corollary 5.2.1.** If $char(\mathbb{F}) = p$, and $|\mathbb{F}| = q$, then $|\mathcal{P}_{\mathbf{d},r}| = \min(q^{k \times n}, \sum_{i=1}^{m} N_i(n, k) + 1)$, where $N_i(n, k)$ is defined in Equation 3.

*Proof.* See Appendix A.10.  □

## 6   CONCLUSIONS

### 6.1   SUMMARY

In this work, we developed an algebraic framework for analyzing shallow polynomial neural networks over finite fields. By formulating expressivity in terms of the cardinality of neuromanifolds, we established general upper bounds, lower bounds, and explicit formulas for a range of architectures. We also highlighted differences between neuromanifolds defined over finite fields and their counterparts over $\mathbb{C}$, particularly in cases where finite field structures constrain the filling property.

### 6.2   FUTURE WORK

A natural extension of our work is to analyze deeper architectures (with multiple hidden layers) to investigate how composition of several layers affects the size and geometry of the neuromanifolds. Another is to study more general polynomial activations.

The projective cardinalities introduced and computed in Section 4 can also be combined with the Weil conjectures to compute the cohomology of complex neuromanifolds, thereby providing new insights into their topological structure.

In addition, we have identified several open problems throughout our main sections that merit further examination, such as the question of simultaneous diagonalization of tensors.

### 6.3   CONCLUSION

Our results provide evidence that recent algebraic approaches can play a meaningful role in the study of neural networks, complementing the more familiar statistical and probabilistic perspectives.

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

## A APPENDIX

### A.1 PROOF OF PROPOSITION 2.2

*Proof.* Write $r = \sum_{t=0}^{s_r} r_t\, p^t$ with digits $0 \le r_t \le p-1$. For each exponent tuple $\alpha = (\alpha_1, \ldots, \alpha_n)$ with $\sum_i \alpha_i = r$ let $\alpha_{i,t} \in \{0, \ldots, p-1\}$ be its $t$-th base-$p$ digit so that $\alpha_i = \sum_t \alpha_{i,t} p^t$.

By Theorem A.1, the multinomial coefficient $\binom{r}{\alpha_1, \ldots, \alpha_n}$ is non-zero mod $p$ exactly when every one-digit factor $\binom{r_t}{\alpha_{1,t}, \ldots, \alpha_{n,t}}$ is non-zero. Since each $r_t < p$, none of the factorials occurring in that one-digit factor is divisible by $p$, so the only condition is the sum constraint $\alpha_{1,t} + \cdots + \alpha_{n,t} = r_t$.

For fixed $t$, the number of $(\alpha_{1,t}, \ldots, \alpha_{n,t})$ satisfying that constraint equals the stars-and-bars count $\binom{r_t+n-1}{r_t}$. Digit positions are independent, hence the total number of exponent tuples that give non-vanishing coefficients is the product of those counts:

$$\gamma_{n,p}(r) = \prod_{t=0}^{s_r} \binom{n + r_t - 1}{r_t},$$

as claimed.

Naturally, this means that $\mathcal{P}_{(n,m,1),r} \le \gamma_{n,p}(r)$. It is not difficult to extend to $k > 1$. Consider the $k-$tuple of outputs $(u_1, u_2, \ldots, u_k)$: each $u_i$ has at most $p^{\gamma_{n,p}(r)}$ choices. Since the outputs are independent, we get $\mathcal{P}_{(n,m,k),r} \le p^{\gamma_{n,p}(r) \cdot k}$. $\qquad\square$

**Theorem A.1** (Multinomial Congruence, (Conrad, 1995)). *Let $p$ be a prime. Fix integers $d \ge 0$ and $t \ge 1$. Suppose $s_0, s_1, \ldots, s_t \ge 0$. Write*

$$s_0 = c_0 + c_1 p + \cdots + c_d p^d, \qquad 0 \le c_i \le p-1 \ (i < d),$$

$$s_j = c_{0j} + c_{1j} p + \cdots + c_{dj} p^d, \qquad 0 \le c_{ij} \le p-1 \ (i < d, \ 1 \le j \le t),$$

*with $c_d, c_{dj} \ge 0$. Then*

$$\binom{s_0}{s_1, \ldots, s_t} \equiv \binom{c_0}{c_{01}, \ldots, c_{0t}} \binom{c_1}{c_{11}, \ldots, c_{1t}} \cdots \binom{c_d}{c_{d1}, \ldots, c_{dt}} \pmod{p}.$$

### A.2 SDC ALGORITHM AND PROOF OF THEOREM 3.5

We will use the following corollary of Bustamante et al. (2020, Theorem 7).

**Lemma A.2.** *Let $A_1, A_2$ be $n \times n$ symmetric matrices, where $A_1$ is full-rank. Then, $A_1, A_2$ are simultaneously diagonalizable via congruence if and only if there exists a diagonal matrix $D$ and an invertible $n \times n$ matrix $Q$ such that*

$$A_1^{-1} A_2 = Q D Q^{-1}.$$

*Proof of Theorem 3.5.* We first give an example of a pair of matrices which are *not* simultaneously diagonalizable via congruence (SDC) over $\mathbb{C}$. Take

$$A_1 := \begin{pmatrix} 0 & 1 \\ 1 & 0 \end{pmatrix}, \quad A_2 := \begin{pmatrix} 0 & 0 \\ 0 & 1 \end{pmatrix}.$$

If we assume, for the sake of contradiction, there is some invertible $Q = \begin{pmatrix} a & b \\ c & d \end{pmatrix}$ which simultaneously diagonalizes $A_1$ and $A_2$, then

$$Q A_2 Q^T = \begin{pmatrix} a & b \\ c & d \end{pmatrix} \begin{pmatrix} 0 & 0 \\ 0 & 1 \end{pmatrix} \begin{pmatrix} a & c \\ b & d \end{pmatrix} = \begin{pmatrix} b^2 & bd \\ bd & d^2 \end{pmatrix}$$

implies $b$ or $d = 0$. Meanwhile,

$$Q A_1 Q^T = \begin{pmatrix} a & b \\ c & d \end{pmatrix} \begin{pmatrix} 0 & 1 \\ 1 & 0 \end{pmatrix} \begin{pmatrix} a & c \\ b & d \end{pmatrix} = \begin{pmatrix} b & a \\ d & c \end{pmatrix} \begin{pmatrix} 2ab & bc + ad \\ bc + ad & 2dc \end{pmatrix}$$

implies $bc + ad = 0$. Without loss of generality, if $b = 0$, then $a$ or $d = 0$. Either way, $Q$ would have a row or column of zeros, and therefore $Q$ would be non-invertible, a contradiction.

Alternatively, we may remark $A_1^{-1}A_2 = \begin{pmatrix} 0 & -1 \\ 0 & 0 \end{pmatrix}$, is a Jordan block with eigenvalues 0, which is not diagonalizable, and thus $A_1$, $A_2$ are not SDC by the above lemma.

Therefore, the neuromanifold $\mathcal{P}_{(n,n,2),2}$ over $\mathbb{C}$ is not filling.

However, we still wish to show the Zariski closure of $\mathcal{P}_{(n,n,2),r}$ over $\mathbb{C}$ is the entire ambient space $S_2(\mathbb{C}^n) \times S_2(\mathbb{C}^n)$, for which we apply Lemma A.2.

If $A_1$ is full rank and $A_1^{-1}A_2$ has no repeat eigenvalues, then its Jordan normal form is diagonal. That is, there will exist a change of basis matrix $Q$ such that $A_1^{-1}A_2 = QDQ^{-1}$, and $A_1$, $A_2$ will be SDC by our lemma.

$B := A_1^{-1}A_2$ has no repeat eigenvalues if and only if the discriminant $\Delta$ of its characteristic polynomial $f(x) = \text{charpoly}(B) = \det(B - Ix)$ is non-zero. The discriminant of a polynomial can be expressed as the *resultant* of the polynomial $f(x)$ and its derivative $f'(x)$. Notably, $f(x)$ and $f'(x)$ are polynomials whose coefficients are polynomials in the entries of $A_1$, $A_2$, and the resultant is a polynomial in said coefficients. Hence, the resultant is a polynomial in the entries of $A_1$, $A_2$, and thus

$$V_1 : \quad \Delta(\text{charpoly}(B)) = 0$$

is an algebraic variety in the entries of $A_1$, $A_2$.

Meanwhile, the set $\{\det A_1 = \det A_2 = 0\}$ is also an algebraic variety. Specifically, $\{\det A_1 = \det A_2 = 0\}$ is the variety

$$V_2 := \cap_{\substack{i \in \{1,2\} \\ 1 \le j,k \le n}} V_{i,j,k}$$

where

$$V_{i,j,k} := \{\det(A_i)_{j,k} = 0\}$$

is the variety given by the determinant of the $j, k$-th minor of $A_i$. We then have $(S_2(\mathbb{C}^n))^2 \setminus (V_1 \cup V_2)$ as a subset of all 2-tuples $A_1$, $A_2$ which are SDC. Thus,

$$S_2(\mathbb{C}^n) \setminus (V_1 \cup V_2) \subseteq \mathcal{P}_{(n,n,2),2}.$$

$V_1 \cup V_2$ is a proper Zariski-closed subset of $(S_2(\mathbb{C}^n))^2 \cong \mathbb{C}^{(n+1)n}$. Hence, its complement $(S_2(\mathbb{C}^n))^2 \setminus (V_1 \cup V_2)$ is a non-empty Zariski-open set. Non-empty Zariski-open sets are dense in $\mathbb{C}^k$ for any natural number $k$. Therefore, the Zariski closure of $(S_2(\mathbb{C}^n))^2 \setminus (V_1 \cup V_2)$ (and hence of the superset $\mathcal{P}$) is the entire ambient space $(S_2(\mathbb{C}^n))^2$, as desired. $\qquad\square$

### A.3 PROOF OF LEMMA 4.1

*Proof.* The $k$-tuple consisting of all 0-matrices lies in $\mathcal{P}_{\mathbf{d},r}$. For all remaining $k$-tuples $(A_1, \ldots, A_k) \in \mathcal{P}_{\mathbf{d},r}$, scaling by any $\lambda \in \mathbb{F}_q \setminus \{0\}$ gives another point $\lambda(A_1, \ldots, A_k) \in \mathcal{P}_{\mathbf{d},r}$, distinct if $\lambda \ne 1$. Therefore,

$$|\overline{\mathcal{P}_{\mathbf{d},r}}| = \frac{|\mathcal{P}_{\mathbf{d},r}| - 1}{q - 1}.$$

Rearranging gives the desired result. $\qquad\square$

### A.4 PROOF OF LEMMA 4.2

*Proof.* The elements of $\overline{\mathcal{P}_{\mathbf{d},r}}$ are $k$-tuples of symmetric tensors $[A_1 : \ldots : A_k]$ up to scaling, where

$$A_i = b_i L^{\otimes r}.$$

If we fix $L$ up to scaling, then each $A_i$ lies on the line in $S_r(\mathbb{F}_q^n)$ generated by $L^{\otimes r}$. We claim different choices of $[L]$ (choices of $L$ up to scaling) partition $\overline{\mathcal{P}_{\mathbf{d},r}}$. In other words, we claim given

distinct $[L_1]$, $[L_2] \in \mathbb{P}(\mathbb{F}_q^n) \cong \mathbb{P}^{n-1}$ over $\mathbb{F}_q$, the resulting $L_1^{\otimes r}$ and $L_2^{\otimes r}$ define different lines in $S_r(\mathbb{F}_q^n)$ (alternatively stated, $L_1^{\otimes r}$ and $L_2^{\otimes r}$ are not scalar multiples of each other).

We can assume by change of basis that $L_1 = e_1$ is a standard basis vector. Then, $L_1^{\otimes r}$ is an order-$r$ symmetric tensor with a 1 at the entry indexed by $(1, \ldots, 1)$, and zero everywhere else. If $L_2 = \lambda L_1$ where $\lambda \neq 0$, then the first entry of $L_2$, denoted $(L_2)_1$, must be non-zero. Meanwhile, the $j$-th entry of $L_2$, denoted $(L_2)_j$, must be zero, since there exists an entry corresponding to $(L_2)_1^{r-1}(L_2)_j = 0$. Therefore, $L_2$ is a multiple of $e_1 = L_1$. Therefore, if $\left[L_1^{\otimes r}\right] = \left[L_2^{\otimes r}\right]$ in $\mathbb{P}(S_r(\mathbb{F}_q^n))$, then $[L_1] = [L_2]$.

So, choice of $[L]$ partition $\overline{\mathcal{P}_{\mathbf{d},r}}$. We furthermore observe, given a choice of $[L] \in \mathbb{P}(\mathbb{F}_q^n) \cong \mathbb{P}^{n-1}$, $[b_1 : ... : b_k] \in \mathbb{P}^{k-1}$ determines $[A_1 : ... : A_k]$ uniquely. Thus,

$$\overline{\mathcal{P}_{\mathbf{d},r}} \cong \mathbb{P}^{n-1} \times \mathbb{P}^{k-1}.$$

By removing the origin and dividing by $q - 1$ to account for scalars,

$$|\mathbb{P}^{i-1}| = \frac{|\mathbb{F}_q^i| - 1}{q - 1} = \frac{q^i - 1}{q - 1}$$

for all $i \in \mathbb{N}$. Therefore,

$$|\overline{\mathcal{P}_{\mathbf{d},r}}| = |\mathbb{P}^{n-1}||\mathbb{P}^{k-1}| = \frac{(q^n - 1)(q^k - 1)}{(q - 1)^2}.$$

$\square$

### A.5 Proof of Proposition 4.3

*Proof.* The elements of $\overline{\mathcal{P}_{\mathbf{d},r}}$ are projective equivalence classes of ordered $k$-tuples of $n^{\oplus r}$ symmetric tensors $[A_1 : ... : A_k]$ of the form

$$A_i = b_{i1}L_1^{\otimes r} + b_{i2}L_2^{\otimes r}$$

where $\left[L_1^{\otimes r}\right], \left[L_2^{\otimes r}\right] \in \overline{\mathcal{P}_{(n,1,1),r}}$ are equivalence classes of rank-1 $n^{\otimes r}$ symmetric tensors.

We consider $L_1^{\otimes r}$, $L_2^{\otimes r}$ as elements of a vector space over $\mathbb{F}_q$. Then, since we have two basis elements $L_1^{\otimes r}$ and $L_2^{\otimes r}$, $[A_1 : ... : A_k]$ spans 1 or 2 dimensions.

If the tensors in $[A_1 : ... : A_k]$ span 1 dimension, then, in fact, all $A_i$ are of the form $A_i = b_i A$ for some $A \in \mathcal{P}_{(n,2,1),r}$. The choice of $[A]$ and parameterization $[B] = [b_1, b_2, ..., b_k] \in \mathbb{P}^{k-1}$ uniquely determines $[A_1 : ... : A_k]$. Hence,

$$\{\text{1-dimensional } [A_1 : ... : A_k]\} \cong \overline{\mathcal{P}_{(n,2,1),r}} \times \mathbb{P}^{k-1}$$

and we obtain the term

$$\left|\overline{\mathcal{P}_{(n,2,1),r}}\right| \left|\mathbb{P}^{k-1}\right|.$$

In the case of a 2-dimensional span by $[A_1 : ... : A_k]$, we begin by arguing the choice of basis pair $\left[L_1^{\otimes r}\right], \left[L_2^{\otimes r}\right]$ partitions the set of 2-dimensional $[A_1 : ... : A_k]$ disjointly.

For the sake of contradiction, suppose some 2-dimensional $[A_1 : ... : A_k]$ may be constructed by two different basis pairs $\left[L_1^{\otimes r}\right], \left[L_2^{\otimes r}\right]$ and $\left[L_3^{\otimes r}\right], \left[L_4^{\otimes r}\right]$. Since $[A_1 : ... : A_k]$ is 2-dimensional, the $A_i$ form an alternative basis for the space spanned by $L_1^{\otimes r}$ and $L_2^{\otimes r}$ (or $L_3^{\otimes r}, L_4^{\otimes r}$) and we can therefore recover the original basis $L_1^{\otimes r}, L_2^{\otimes r}$ and $L_3^{\otimes r}, L_4^{\otimes r}$ as a linear combination of the $A_i$'s. Consequently, $L_3^{\otimes r}, L_4^{\otimes r}$ are both in the span of $L_1^{\otimes r}, L_2^{\otimes r}$, and the two pairs form the same plane in $S_r^{(}\mathbb{F}_q^n)$. Note, for the pairs to be distinct, at least one of these linear combinations of $L_1^{\otimes r}, L_2^{\otimes r}$ is non-trivial. More precisely, there exists $c_1, c_2 \neq 0$ such that

$$L_j^{\otimes r} = c_1 L_1^{\otimes r} + c_2 L_2^{\otimes r}$$

where $j = 3$ or $4$.

Since $\left[L_1^{\otimes r}\right] \neq \left[L_2^{\otimes r}\right]$, therefore $L_1^{\otimes r}, L_2^{\otimes r}$ are linearly independent, and consequently so are $L_1$ and $L_2$. If we apply a change of basis such that $L_1 = (1, 0, 0, ..., 0)$ and $L_2 = (0, 1, 0, ..., 0)$ are standard basis vectors, then $c_1 L_1^{\otimes r} + c_2 L_2^{\otimes r}$ is an $n^{\otimes r}$ tensor with one $c_1$ and one $c_2$ along the diagonal and 0's everywhere else. Since $r > 1$, therefore this tensor is of symmetric CP-rank rank 2. In other words, there does not exist an $L_3$ such that $L_3^{\otimes r} = c_1 L_1^{\otimes r} + c_2 L_2^{\otimes r}$ where $c_1, c_2 \neq 0$. Therefore, we have a contradiction, and in fact, our choice of distinct $\left[L_1^{\otimes r}\right], \left[L_2^{\otimes r}\right]$ will partition the set of 2-dimensional $[A_1 : ... : A_k]$'s disjointly.

Fixing a choice of distinct $\left[L_1^{\otimes r}\right]$ and $\left[L_2^{\otimes r}\right] \in \overline{\mathcal{P}_{(n,1,1),r}}$, $[A_1 : ... : A_k]$ is 2-dimensional if and only if the coefficients as vectors $\begin{pmatrix} b_{i1} \\ b_{i2} \end{pmatrix}$ span two dimensions. That is, each possible $[A_1 : ... : A_k]$ is uniquely parameterized by a choice of basis $\left[L_1^{\otimes r}\right], \left[L_2^{\otimes r}\right] \in \overline{\mathcal{P}_{(n,1,1),r}}$ and some rank-2 matrix

$$[(B)] = \left[ \begin{pmatrix} b_11 & b_21 & \ldots & b_k1 \\ b_12 & b_22 & \ldots & b_k2 \end{pmatrix}^T \right] \in \mathbb{P}(M^{2 \times k}).$$

There are $\binom{\left|\overline{\mathcal{P}_{(n,1,1),r}}\right|}{2}$ ways to choose $\left[L_1^{\otimes r}\right], \left[L_2^{\otimes r}\right]$. Fixing $\left[L_1^{\otimes r}\right], \left[L_2^{\otimes r}\right]$, the number of choices of matrix projective point $[(B)]$ is $\frac{N_2(2,k)}{q-1} = \frac{(q^k-1)(q^k-q)}{q-1}$. Hence, we obtain our second term, the count of 2-dimensional $[A_1 : ... : A_k]$,

$$\binom{\left|\overline{\mathcal{P}_{(n,1,1),r}}\right|}{2} \frac{N_2(2, k)}{q - 1}.$$

Therefore,

$$\left|\overline{\mathcal{P}_{\mathbf{d},r}}\right| = \underbrace{\left|\overline{\mathcal{P}_{(n,2,1),r}}\right| \cdot \left|\mathbb{P}^{k-1}\right|}_{\text{1-dimensional span}} + \underbrace{\binom{\left|\overline{\mathcal{P}_{(n,1,1),r}}\right|}{2} \cdot \frac{N_2(2, k)}{q - 1}}_{\text{2-dimensional span}}.$$

as desired. $\qquad \square$

### A.6    PROOF OF PROPOSITION 4.4

*Proof.* Compared to the previous proposition, the term corresponding to $k$-tuples $[A_1 : \ldots : A_k]$ with a 1-dimensional span remains unchanged.

However, the contribution from $k$-tuples $[A_1 : \ldots : A_k]$ with 2-dimensional span is divided by $\frac{q(q+1)}{2}$. This factor arises because, when $r = 1$, linear combinations of $L_1^{\otimes r}, L_2^{\otimes r}$ are no longer rank 2, but rank 1. More specifically, we have

$$c_1 L_1^{\otimes r} + c_2 L_2^{\otimes r} = c_1 L_1 + c_2 L_2 = L_3 = L_3^{\otimes r}.$$

so any linear combination is itself another vector $L_3 \in F_q^n$.

Consequently, the choice of basis pair $[L_1], [L_2]$ corresponding to the plane spanned by $[A_1 : \ldots : A_k]$ is no longer unique, unlike in the case $r > 1$ considered in Proposition 4.3. For instance, one could take $L_3 = L_1 + L_2$ and $L_1$ as the spanning pair instead. In fact, any two linearly independent elements of the plane will suffice as a basis, since with $r = 1$, each tensor in the plane is a vector.

The number of unordered pairs of linearly independent elements in a 2-dimensional subspace over $\mathbb{F}_q$ is

$$\frac{(q^2 - 1)(q^2 - q)}{2}.$$

After projectivising (i.e., accounting for scaling by a non-zero scalar), we have

$$\frac{(q^2 - 1)(q^2 - q)}{2} \cdot \frac{1}{(q - 1)^2} = \frac{q(q + 1)}{2}.$$

Hence, each 2-dimensional $[A_1 : \ldots : A_k]$ can be attained by $\frac{q(q+1)}{2}$ distinct basis pairs $\left[L_1^{\otimes r}\right], \left[L_2^{\otimes r}\right]$. Therefore, we divide by this factor to adjust for overcounting. $\qquad \square$

## A.7 Proof of Proposition 5.1

*Proof.* As we defined above, let $a_{ij}$ be the $(i,j)$th entry of $W_1$ and $b_{ij}$ be the $(i,j)$th entry of $W_2$. Direct computation shows that $k$th entry of $\sigma_1 W_1(\mathbf{x})$ is equal to

$$(\sigma_1 W_1(\mathbf{x}))_k := \left( \sum_{i=1}^{n} a_{ki} x_i \right)^{ip}$$

Since $p$ is prime, then we have the following identity

$$(a+b)^p \equiv a^p + b^p \text{ for all } a, b \in \mathbb{F}.$$

Computing the $k$th term, for example, gives that

$$\left( \sum_{i=1}^{n} a_{ki} x_i \right)^p = \left( a_{k1} x_1 + \sum_{i=2}^{n} a_{ki} x_i \right)^p = (a_{k1} x_1)^p + \left( \sum_{i=2}^{n} a_{ki} x_i \right)^p$$

If we now recursively simplify the last term, we will eventually get that:

$$\left( \sum_{i=1}^{n} a_{ki} x_i \right)^p = \sum_{i=1}^{n} a_{ki}^p x_i^p$$

Thus,

$$\left( \sum_{i=1}^{n} a_{ki} x_i \right)^{ip} = \left( \sum_{i=1}^{n} a_{ki}^p x_i^p \right)^i$$

The same computation done with $r = i$ will give us the same expression but with $a_{ki} x_i$ replacing the $a_{ki}^p x_i^p$. It is immediate that the number of polynomials in $\mathcal{P}_{\mathbf{d}, ip}$ is at most the number of polynomials in $\mathcal{P}_{\mathbf{d}, i}$ since $a_{ki}^p \in \mathbb{F}$ for all any $a_{ki} \in \mathbb{F}$. Thus, to show that $|\mathcal{P}_{\mathbf{d}, ip}| = |\mathcal{P}_{\mathbf{d}, i}|$, it suffices to prove that every element of $\mathbb{F}$ is a $p$th power, to allow the coefficient before the $x_i^p$ term to be any element from $\mathbb{F}$. Fortunately, it is a standard result in algebra that all finite fields are perfect (**?**, Theorem 3), meaning that for all $b \in \mathbb{F}$, there exists $a \in \mathbb{F}$ such that $b = a^p$, if $char(\mathbb{F}) = p$. Now consider arbitrary $b_{ki} \in \mathbb{F}$; we can choose $a_{ki} \in \mathbb{F}$ such that $b_{ki} = a_{ki}^p$. Let $y_i$ denote $x_i^p$. Then we have that

$$(\sigma_1 W_1(\mathbf{x}))_k = \left( \sum_{i=1}^{n} b_{ki} y_i \right)^i$$

Now we observe that this is exactly the same as $(\sigma_1 W_1(\mathbf{x}))_k$ when computed with $r = i$ instead. Since all polynomials are in the form of $W_2(\sigma_1(W_1(\mathbf{x})))$, and $W_2$ does not depend on $r$, we can conclude that $|\mathcal{P}_{\mathbf{d}, ip}| = |\mathcal{P}_{\mathbf{d}, i}|$. $\qquad \square$

## A.8 Proof of Corollary 5.1.1

*Proof.* We prove it by induction on $i$; suppose $i = 1$, then by Proposition 5.1,

$$|\mathcal{P}_{\mathbf{d}, r}| = |\mathcal{P}_{\mathbf{d}, ps}| = |\mathcal{P}_{\mathbf{d}, s}|$$

Suppose

$$|\mathcal{P}_{\mathbf{d}, p^{i-1}s}| = |\mathcal{P}_{\mathbf{d}, s}|$$

then by Proposition 5.1,

$$|\mathcal{P}_{\mathbf{d}, r}| = |\mathcal{P}_{\mathbf{d}, p^i s}| = |\mathcal{P}_{\mathbf{d}, p^{i-1}s}| = |\mathcal{P}_{\mathbf{d}, s}|$$

$$\square$$

## A.9 Proof of Proposition 5.2

Before we begin the proof, we first prove a short lemma that will be used:

**Lemma A.3.** *A matrix $C \in \mathbb{F}^{k \times n}$ is representable as a product of two matrices matrices $B \times A$, where $B \in \mathbb{F}^{k \times m}$, $A \in \mathbb{F}^{m \times n}$ if and only if $C$ has rank $\leq m$.*

*Proof.* Consider $i$ th column of $C$ for some fixed $i$. Then $C_i = B \times A_i$, where $A_i$ is the $i$th column of $A$.

For the forward direction, from the assumptions, the $i$th column of $C$ is a linear combination of the columns of $B$, with the coefficients determined by the $i$th column of $A$ (for all $1 \leq i \leq n$). Thus, the column span of $C$ can be spanned by a collection of $m$ vectors (since $B$ has $m$ columns), meaning that $C$ has column rank $\leq m$.

Now suppose $C \in \mathbb{F}^{k \times n}$ has rank $\leq m$; then there exists a collection of $m$ vectors in $\mathbb{F}^k$ such that all columns of $C$ are linear combinations of vectors from the collection; then choose the columns of $B$ to be the vectors in the collection, and the entries of the $i$th column of $A$ to be the coefficients in linear combination for the $i$th column of $C$, which will give us $C = B \times A$ □

Now we start the actual proof of Corollary 5.2

*Proof.* By Corollary 5.1.1, we have that for all $i \in \mathbb{N}$, $|\mathcal{P}_{\mathbf{d},p^i}| = |\mathcal{P}_{\mathbf{d},1}|$. Now we need to compute the cardinality of $\mathcal{P}_{\mathbf{d},1}$, where the network output is given by

$$f_{\mathbf{w}}(\mathbf{x}) = W_2 \sigma_1 W_1 \mathbf{x} = W_2 W_1 \mathbf{x}.$$

By direct computations, we see that in order to learn $f(\mathbf{x}) = C\mathbf{x}$, we must have the following decomposition:

$$C = W_2 \times W_1.$$

where $C \in \mathbb{F}^{k \times n}, W_2 \in \mathbb{F}^{k \times m}$, and $W_1 \in \mathbb{F}^{m \times n}$. Thus, according to Lemma A.3, the neuromanifold exactly corresponds to the set of matrices with rank at most $m$. □

## A.10 Proof of Corollary 5.2.1

*Proof.* By Proposition 5.2, we have that $|\mathcal{P}_{\mathbf{d},r}|$ is the number of all $k \times n$ matrices with rank $\leq m$.

If $m \geq \min(n, k)$, then since all matrices of size $k \times n$ has rank at most $\min(k, n)$, and $m \geq \min(n, k)$, all $k \times n$ matrices satisfy the condition of having rank at most $m$. Thus, by Proposition 5.2, $|\mathcal{P}_{\mathbf{d},r}| = q^{k \times n}$.

By Equation 3, we have that $N_i(n, k)$ is the number of $k \times n$ matrices with rank exactly $i$. Thus, if $m < \min(n, k)$, $|\mathcal{P}_{\mathbf{d},r}| = \sum_{i=1}^{m} N_i(n, k) + 1$, where the 1 corresponds to the zero matrix, which has rank 0. And this expression is indeed smaller than $p^{k \times n}$ since the latter expression also includes matrices with rank $> m$ but $\leq \min(k, n)$

□

