# OpenReview forum: "Expressivity of Shallow Neural Networks Over Finite Fields"
_ICLR.cc/2026/Conference — Submitted to ICLR 2026_

### Official Review · Reviewer_AJyj · 2025-10-15

**Soundness:** 3
**Presentation:** 3
**Contribution:** 2
**Rating:** 4
**Confidence:** 4

**Summary:**

This paper analyzes the expressivity of shallow (one hidden-layer) monomial MLPs over finite fields. Over characteristic 0, expressivity is commonly quantified by the dimension of the corresponding neuromanifold (the image of the parameterization). In the finite-field setting, the authors instead define expressivity by a counting ratio $|P_{\mathbf{d},r}|/|(S^r(\mathbb{F}^n))^k|$, i.e., the number of realizable outputs (the neuromanifold $P_{\mathbf{d},r}$
 ) divided by the number of all $k$-tuples of homogeneous degree-$r$ polynomials in $n$ variables. They derive exact formulas and upper bounds on $|P_{\mathbf{d},r}|$, which are summarized in Table 1.

**Strengths:**

The paper proposes an elegant approach to measuring expressivity, making use of powerful tools from number theory and algebraic geometry. The results are clearly presented and the proofs are sound. While there are some restrictions (such as the choice of activation and architecture) and natural doubts about the potential of this line of research, new approaches should be welcomed, especially when they are presented as clearly as in this work.

**Weaknesses:**

The main weakness of this paper is that it only considers shallow networks without bias and with monomial activation $x^r$. An analysis using more general activation functions could be of greater interest. In addition, while the proposed measure of expressivity -- defined as a ratio -- makes sense in the shallow setting, it may become less informative for deeper networks, where the ambient space contains vastly more polynomials and the ratio therefore becomes very small.

**Questions:**

I am particularly curious about the analysis of identifiability in this setting. When $p$ is sufficiently large in $\mathbb{F}_p$, is it possible to apply techniques from the Waring rank problem (as in Finkel et al., 2025) to describe most of the fibers? (I avoid saying "generic fiber" here since we are working over finite fields.) If so, could this yield an upper bound based on counting the fibers? Perhaps such an approach could also extend to networks of arbitrary depth.


As for my last question, do you see a possible resolution when considering $p$-adic fields? It might be highly relevant if one could take this approach and compare the expressivity in that setting.

---

### Official Review · Reviewer_V6bH · 2025-10-31

**Soundness:** 2
**Presentation:** 1
**Contribution:** 3
**Rating:** 6
**Confidence:** 2

**Summary:**

The paper studies the expressive power of shallow polynomial neural networks over finite fields, motivated by quantization and by links between rational point counting and the Weil conjectures. Expressivity is framed geometrically via neuromanifolds, defined as the image of the parameter map that sends weights to tuples of homogeneous degree-(r) polynomials. Because these ambient polynomial spaces are finite, expressivity is quantified by the cardinality ratio $|P_{d,r}| / |(S_r(\mathbb{F}^n))^k|$. The work extends prior analyses over $\mathbb{R}$ and $\mathbb{C}$ to finite fields, develops a general upper bound using a multinomial congruence, gives exact counts for special architectures, and contrasts finite-field behavior with characteristic-zero using Zariski closures. A notable phenomenon is that a neuromanifold can fill an open dense set over the complex numbers yet occupy only about half of the ambient space over finite fields. The authors also show that when the field characteristic divides the activation degree, the count is unchanged by a Frobenius argument, and they close with open problems and a path from point counts to cohomology via Weil’s conjectures.

**Strengths:**

The paper is well motivated by practical quantization and low-precision arithmetic, and by the conceptual bridge from point counts to topological information through the Weil conjectures. It offers a clear algebraic and counting-based framework that defines neuromanifolds and measures expressivity by a finite cardinality ratio. The authors combine algebraic geometry, finite field linear algebra, and combinatorics to derive a general upper bound, compute exact cardinalities for several architectures, and highlight sharp differences between finite-field and characteristic-zero behavior through Zariski closures. The result that finite-field neuromanifolds can behave very differently from their real or complex analogs, even for shallow networks, is compelling. The Frobenius observation that characteristic dividing the activation degree leaves the count unchanged strengthens the conceptual message, and the roadmap to connect point counts to cohomology rounds out the contribution.

**Weaknesses:**

Given my understanding of the field, it’s hard to accurately evaluate the technical contribution of this paper. However, I would like to raise the following two points.

The Related Work omits several directly relevant strands that would situate the contribution more clearly. First, there is a mature line on depth versus width for polynomial targets in real-valued networks that motivates why staying shallow is a strong limitation, including explicit exponential gaps that your finite-field setting could mirror or contrast. Second, because a one-hidden-layer network with activation x^r) outputs sums of (r)-th powers of linear forms, the architecture coincides with the classical Waring model for homogeneous polynomials, so Waring-rank bounds are the natural structural proxy for required width. See for example these three papers [1], [2] and [3].

[1] Shapira, Expressivity of Shallow and Deep Neural Networks for Polynomial Approximation
[2] Carlini–Catalisano–Geramita, The solution to the Waring problem for monomials
[3] Daniely, Depth Separation for Neural Networks and Telgarsky, Benefits of Depth in Neural Networks

The paper contains frequent spelling and grammar mistakes (e.g. lines 267, 348, 141), sloppy citations and substantial notation inconsistencies, which hinder readability and validation of the results. Several core statements also require correction or clarification, including the parameter count in Definition 2.1, the affine versus projective notation in Section 4, the units in the upper bound, and the numbers in Example 3.6.

**Questions:**

Please address my concerns above

---

### Official Review · Reviewer_UQ6b · 2025-10-31

**Soundness:** 3
**Presentation:** 2
**Contribution:** 2
**Rating:** 2
**Confidence:** 4

**Summary:**

This paper introduces a new measure of expressivity for polynomial neural networks by taking advantage of the algebro-geometric structure of such models.  It shows that the mathematical notion of field characteristic has important implications on the proposed definition of expressivity.

**Strengths:**

The writing overall is good.  There are some quite technical mathematical concepts and a good effort was made to make them accessible to the general machine learning audience, but I still feel that this submission is mainly of interest to mathematicians.

**Weaknesses:**

Quite minor but I find the use of subsections to be excessive.  More importantly, the lack of implementation and relation to practicality are major flaws.

**Questions:**

This is not so much a question, but rather an admittedly skeptical observation, which is why I put it here rather than the "Weaknesses" section: PNNs are interesting mathematical objects but they are really limited in practical utility (notably due to their very poor scalability).  It feels to me like the new literature, including the current submission, is largely a well-defined algebro-geometric setting where tools from this theory can be readily applied and nice results (including those given in the submission) can be achieved, with little regard to actual practicality.  Similarly, shallow neural networks are useful in limited settings, but provide a very clean algebraic setting for theoretical (mathematical, rather than machine learning) questions.  Although I agree that they're still interesting objects and settings to study from the point of view of the mathematician, my view is that shallow PNNs are of little interest to machine learning practitioners, which is largely the audience of conferences such as ICLR.  It is my view that such a submission would be better suited to a "mathematics of machine learning" journal rather than a conference such as ICLR.  I think the result is interesting, mathematically, but not for the ICLR audience.

---

### Official Review · Reviewer_FWBX · 2025-11-01

**Soundness:** 3
**Presentation:** 2
**Contribution:** 3
**Rating:** 6
**Confidence:** 2

**Summary:**

This paper conducts a pioneering theoretical study on the expressive power of shallow polynomial neural networks (PNNs) over the finite field $\mathbb{F}_q$. The authors define expressive power as the cardinality of their "neural manifold" and successfully provide exact formulas for this cardinality for several key classes of architectures (e.g., $r=2, k=1$ and $m=1,2$). In particular, the authors leverage the algebraic properties of the finite field's characteristic $p$ (when $p | r$) to simplify the problem. A core contribution is the revelation, through a key example ($d=(2,2,2), r=2$), of the significant difference in expressive power between finite fields and the complex numbers, showing that an architecture that "fills" the space over $\mathbb{C}$ may only occupy a small fraction of the ambient space over $\mathbb{F}_q$.

**Strengths:**

The main strength of this paper lies in its novelty. It provides a rigorous, algebraic-geometric framework for analysis for the important problem of "quantifying the expressive power of neural networks" (i.e., calculating the cardinality of the neural manifold over $\mathbb{F}_q$). The paper not only proposes a framework but also delivers very solid and concrete mathematical results, providing exact cardinality counting formulas for many classes of architectures (as shown in Table 1). The authors skillfully apply advanced mathematical tools (such as projective geometry and finite field algebra) to solve complex counting problems in an elegant manner, for example, leveraging the property $(a+b)^p = a^p + b^p$ to simplify the case where $p|r$. A valuable contribution of this paper is its profound insight: it strongly demonstrates that the expressive power over finite fields can be fundamentally different from that over the complex numbers (Zariski closure), which serves as a cautionary note for theoretical research.

**Weaknesses:**

Although theoretical strength, this submission has some notable limitations. The most significant is the weak connection to practice, maybe I am not aware of any direct application at this point.

The paper is motivated by "weight quantization" but does not discuss the specific impact of its mathematical findings (e.g., the neural manifold occupying only half the space in Ex 3.6) on the practical training or generalization ability of quantized networks, which leaves the practical significance of the paper somewhat ambiguous. Furthermore, all analysis is strictly limited to single-hidden-layer (shallow) networks, and it is currently unclear to what extent these conclusions or analytical methods can be generalized to deep networks. At the same time, the authors repeatedly mention a connection to the "Weil Conjectures" (used to infer topological properties of the complex manifold). While this is an exciting motivation, the paper only completes the counting and does not actually demonstrate this connection, making the argument for the motivation incomplete.

**Questions:**

1. could the authors further elaborate on the finding in Ex 3.6 that $\mathcal{P}_{(2,2,2),r=2}$ only "fills" about 1/2 of the ambient space over $\mathbb{F}_p$? Does this imply that the learning capacity of this quantized architecture is severely limited, or could this be seen as a form of "implicit regularization" that is beneficial for generalization?

2. could the authors provide a simple example (even speculatively) to illustrate the connection with the Weil Conjectures? For instance, Lemma 4.2 concludes that $|\overline{\mathcal{P}_{(n,1,k),r}}| = |\mathbb{P}^{n-1}||\mathbb{P}^{k-1}|$. Does this imply that the Betti numbers of its complex manifold are related to those of $\mathbb{P}^{n-1} \times \mathbb{P}^{k-1}$?

3. the authors mention that for the analysis of $m>2$, the "combinatorial complexity grows quickly." Could you specify where this complexity manifests? Is it related to the computation of subspaces of a specific rank (e.g., points on a Grassmannian), or are there more complex combinatorial dependencies?

---

### Meta-Review · Area_Chair_YJwa · 2026-01-07

**Summary:**

As no rebuttal was submitted, the paper is rejected.

**Reviewer Concerns:**

No rebuttal

**Reviewer Scores:**

No rebuttal

---

### Decision · Program_Chairs · 2026-01-26

Reject